# Morphological and Phylogenetic Evidence for Recognition of Two New Species of *Phanerochaete* from East Asia

**DOI:** 10.3390/jof7121063

**Published:** 2021-12-11

**Authors:** Dong-Qiong Wang, Chang-Lin Zhao

**Affiliations:** 1Key Laboratory for Forest Resources Conservation and Utilization in the Southwest Mountains of China, Ministry of Education, Southwest Forestry University, Kunming 650224, China; fungiwangdongqiong@163.com; 2College of Biodiversity Conservation, Southwest Forestry University, Kunming 650224, China; 3Yunnan Key Laboratory for Fungal Diversity and Green Development, Kunming Institute of Botany, Chinese Academy of Sciences, Kunming 650201, China; 4School of Life Sciences, Tsinghua University, Beijing 100084, China

**Keywords:** corticioid fungi, *Phanerochaetaceae*, molecular systematics, taxonomy, Yunnan Province

## Abstract

Two new corticioid fungal species, *Phanerochaete pruinosa* and *P. rhizomorpha* spp. nov. are proposed based on a combination of morphological features and molecular evidence. *Phanerochaete pruinosa* is characterized by the resupinate basidiomata with the pruinose hymenial surface, a monomitic hyphal system with simple-septate generative hyphae and subcylindrical basidiospores measuring as 3.5–6.7 × 1.5–2.7 µm. *Phanerochaete rhizomorpha* is characterized by having a smooth hymenophore covered by orange hymenial surface, the presence of rhizomorphs, subulate cystidia, and narrower ellipsoid to ellipsoid basidiospores. Sequences of ITS+nLSU nrRNA gene regions of the studied specimens were generated and phylogenetic analyses were performed with maximum likelihood, maximum parsimony, and Bayesian inference methods. These phylogenetic analyses showed that two new species clustered into genus *Phanerochaete*, in which *P. pruinosa* was sister to *P. yunnanensis* with high supports (100% BS, 100% BT, 1.00 BPP); morphologically differing by a pale orange to greyish orange and densely cracked hymenial surface. Another species *P. rhizomorpha* was closely grouped with *P. citrinosanguinea* with lower supports; morphologically having yellow to reddish yellow hymenial surface, and smaller cystidia measuring as 31–48 × 2.3–4.8 µm.

## 1. Introduction

Corticioid fungi is a large group of Basidiomycota with simpler basidiomata with the diverse morphological features when compared with polypores, but the phylogenetic diversity of this group is less intensively studied [1,2]. In the subtropical–tropical areas, many corticioid taxa have not been discovered and described worldwide. The genus *Phanerochaete* P. Karst. is a member of the corticioid fungi, which is typified by *P. alnea* (Fr.) P. Karst. [3], and the genus is characterized by the resupinate, membranaceous basidiomata with or without rhizomorphs, a monomitic hyphal system with primarily simple-septate generative hyphae, clavate basidia with four sterigmata, and smooth, thin-walled, inamyloid basidiospores [1,4,5]. Index Fungorum (http://www.indexfungorum.org; accessed on 9 December 2021) registers 187 specific and infraspecific names in *Phanerochaete*. The diversity and taxonomy of *Phanerochaete* s.l. in China have been studied for the last 30 years [6,7,8,9,10,11,12,13,14,15,16,17,18,19].

Molecular studies involving *Phanerochaete* based on the ribosomal DNA (rDNA) sequences, revealed the phylogenetic distribution of resupinate forms across the major clades of mushroom-forming fungi, in which *P. chrysosporium* Burds. nested into phlebioid clade in Polyporales [20]. Revisiting the taxonomy of *Phanerochaete* (Polyporales, Basidiomycota) using a four gene dataset and extensive ITS sampling indicated that *Phanerochaete* sensu lato was polyphyletic and distributed across nine lineages in the phlebioid clade, in which six lineages were associated to described genera [21]. Miettinen et al. [22]. explored the DNA-phylogeny-based and morphology-based to reconcile the polypores and genus concepts in the family *Phanerochaetaceae*, which the macromorphology of fruiting bodies and hymenophore construction did not reflect monophyletic groups, and *Ceriporia inflata* B.S. Jia and B.K. Cui was combined into *Phanerochaete*. Amplifying nrLSU, nrITS, and rpb1 genes across the Polyporales revealed that eleven genera clustered into family *Phanerochaetaceae*, and two families Hapalopilaceae and Bjerkanderaceae were placed as synonyms of *Phanerochaetaceae* [23]. Recently, the research supported by morphological studies and the phylogenetic analyses, showed that many new taxa of *Phanerochaete* s.s. were found and displayed the taxonomic status for the new taxa within genus *Phanerochaete* [14,19].

In 2018–2019, we collected the material supposedly belonging to the two undescribed species of corticioid fungi from Yunnan Province, China. We present the morphological and molecular phylogenetic evidence that support the recognition of two new species within the *Phanerochaete* s.s., based on the internal transcribed spacer (ITS) and regions nLSU sequences.

## 2. Materials and Methods

### 2.1. Morphology

The fruiting bodies were observed growing on the ground of broad-leaved treemixed forest. The fruiting bodies were dried in an electric food dehydrator at 40 °C, then sealed and stored in an envelope bag. They were then transported to Kunming where microscopic morphology and phylogeny to be studied at the mycology laboratory of Southwest Forestry University, Kunming, Yunnan Province, China. The for-study specimens were deposited at the herbarium of Southwest Forestry University (SWFC), Kunming, Yunnan Province, China. Macromorphological descriptions were based on field notes and photos captured in the field and lab. Color terminology follow Petersen [24]. Micromorphological data were obtained from the dried specimens, and observed under a light microscope following Dai [25]. The following abbreviations were used: KOH = 5% potassium hydroxide water solution, CB = Cotton Blue, CB– = acyanophilous, IKI = Melzer’s reagent, IKI– = both inamyloid and indextrinoid, L = mean spore length (arithmetic average for all spores), W = mean spore width (arithmetic average for all spores), Q = variation in the L/W ratios between the specimens studied, *n* = a/b (number of spores (a) measured from given number of specimens (b)).

### 2.2. Molecular Phylogeny

CTAB rapid plant genome extraction kit-DN14 (Aidlab Biotechnologies Co., Ltd., Beijing, China) was used to obtain genomic DNA from dried specimens, according to the manufacturer’s instructions followed previous study [26]. ITS region was amplified with primer pair ITS5 and ITS4 [27]. nLSU region was amplified with primer pair LR0R and LR7 (http://lutzonilab.org/nuclear-ribosomal-dna/; accessed on 28 September 2021). The PCR procedure for ITS was as follows: initial denaturation at 95 °C for 3 min, followed by 35 cycles at 94 °C for 40 s, 58 °C for 45 s, and 72 °C for 1 min, and a final extension of 72 °C for 10 min. The PCR procedure for nLSU was as follows: initial denaturation at 94 °C for 1 min, followed by 35 cycles at 94 °C for 30 s, 48 °C for 1 min and 72 °C for 1.5 min, and a final extension of 72 °C for 10 min. The PCR products were purified and directly sequenced at Kunming Tsingke Biological Technology Limited Company, Kunming, Yunnan Province, China. All newly generated sequences were deposited in NCBI GenBank (Table 1).

Sequences were aligned in MAFFT 7 (https://mafft.cbrc.jp/alignment/server/; accessed on 28 September 2021) using G-INS-i strategy for ITS+nLSU combined dataset, and manually adjusted in BioEdit [38]. Aligned dataset was deposited in TreeBase (submission ID 28442). *Phlebiopsis gigantea* Fr. and *Rhizochaete radicata* (Henn.) Gresl., Nakasone and Rajchenb were selected as an outgroup for phylogenetic analyses of combined dataset following a previous study [19]. The taxon sampling strategy for the selection of sequences for phylogenetic trees was to choose (1) in a larger scale, focusing on the related genera in the families *Phanerochaetaceae* and Irpicaceae in Figure 1; (2) the related taxa based on BLAST search in GenBank within *Phanerochaete* s.l.; and (3) all species of *Phanerochaete* s.s.

Maximum parsimony analysis was applied to the combined (ITS+nLSU) dataset. Its approaches followed Zhao and Wu [26], and the tree construction procedure was performed in PAUP* version 4.0b10 [39]. All characters were equally weighted and gaps were treated as missing data. Trees were inferred using the heuristic search option with TBR branch swapping and 1000 random sequence additions. Max-trees were set to 5000, branches of zero length were collapsed and all parsimonious trees were saved. Clade robustness was assessed using bootstrap (BT) analysis with 1000 replicates [40]. Descriptive tree statistics: tree length (TL), consistency index (CI), retention index (RI), rescaled consistency index (RC), and homoplasy index (HI) were calculated for each Maximum Parsimonious Tree generated. Datamatrix was also analyzed using Maximum Likelihood (ML) approach with RAxML-HPC2 through the CIPRES Science Gateway (www.phylo.org; accessed on 28 September 2021) [41]. Branch support (BS) for ML analysis was determined by 1000 bootstrap replicates.

MrModeltest 2.3 [42] was used to determine the best-fit evolution model for the dataset for Bayesian inference (BI). BI was calculated with MrBayes 3.1.7a [43]. Four Markov chains were run for 2 runs from random starting trees for 10 million generations for ITS+nLSU (Figure 2). The first one-fourth of all generations was discarded as burn-in. The majority rule consensus tree of all remaining trees was calculated. Branches were considered as significantly supported if they received maximum likelihood bootstrap value (BS) >70%, maximum parsimony bootstrap value (BT) >70%, or Bayesian posterior probabilities (BPP) >0.95.

## 3. Results

### 3.1. Molecular Phylogeny

The ITS+nLSU dataset (Figure 1) included sequences from 86 fungal specimens representing 50 species. The dataset had an aligned length of 2368 characters, of which 1170 characters are constant, 598 are variable and parsimony-uninformative, and 600 are parsimony-informative. Maximum parsimony analysis yielded one equally parsimonious tree (TL = 3476, CI = 0.3631, HI = 0.6369, RI = 0.7539, RC = 0.3512). Best model for the ITS+nLSU dataset estimated and applied in the Bayesian analysis was GTR+I+G (lset nst = 6, rates = invgamma; prset statefreqpr = dirichlet (1,1,1,1)). Bayesian analysis and ML analysis resulted in a similar topology to MP analysis with an average standard deviation of split frequencies = 0.038487 (BI), and the effective sample size (ESS) across the two runs is the double of the average ESS (avg ESS) = 303.

The phylogeny (Figure 1) based on the combined ITS+nLSU sequences indicated that both species *Phanerochaete pruinose* and *P. rhizomorpha* clustered into *Phanerochaete* s.s and then *P. pruinose* grouped with *P. subceracea* (Burt) Burds.; *P. rhizomorpha* was sister to *P. citrinosanguinea* Floudas and Hibbett.

The ITS+nLSU dataset (Figure 2) included sequences from 83 fungal specimens representing 53 taxa. The dataset had an aligned length of 2017 characters, of which 1548 characters are constant, 164 are variable and parsimony-uninformative, and 395 are parsimony-informative. Maximum parsimony analysis yielded 35 equally parsimonious trees (TL = 1900, CI = 0.4095, HI = 0.5905, RI = 0.6456, RC = 0.2644). Best model for the ITS+nLSU dataset estimated and applied in the Bayesian analysis was GTR+I+G (lset nst = 6, rates = invgamma; prset statefreqpr = dirichlet (1,1,1,1)). Bayesian analysis and ML analysis resulted in a similar topology to MP analysis with an average standard deviation of split frequencies = 0.004260 (BI), and the effective sample size (ESS) across the two runs is the double of the average ESS (avg ESS) = 309.

The phylogram inferred from ITS+nLSU sequences (Figure 2) revealed that two new species were clustered into genus *Phanerochaete* s.s.; *P. pruinosa* sp. nova. was sister to *P. yunnanensis* Y.L. Xu and S.H. He with high supports (100% BS, 100% BT, 1.00 BPP), and then grouped with *P. robusta* Parmasto without supported data. Another species *P. rhizomorpha* sp. nova. was closely grouped with *P. citrinosanguinea* with lower supports, and then grouped with *P. pseudosanguinea* Floudas and Hibbett (–BS, 98% BP and 1.00 BPP) and *P. sanguinea* (Fr.) Pouzar (96% BS, 91% BP and 1.00 BPP).

### 3.2. Taxonomy

***Phanerochaete pruinosa*** C.L. Zhao and D.Q. Wang, sp. nov. Figure 3 and Figure 4.

MycoBank no.: MB 841271.

Diagnosis: It differs from *P. yunnanensis* by its pruinose hymenophore with the white to slightly cream hymenial surface and lightly darkening in KOH.

**Holotype**—China, Yunnan Province, Chuxiong, Zixishan National Forestry Park, on the bark of fallen angiosperms, 101.4° E, 25.1° N, 1 July 2018, CLZhao 7113 (SWFC).

**Etymology**—***pruinosa*** (Lat.): from Latin, referring to the white powder on hymenial surface of basidiomata.

**Fruiting body**—Basidiomata annual, resupinate, adnate, undetachable from substrate, membranaceous to coriaceous, without odor and taste when fresh, up to 15 cm long, 3 cm wide, 50–100 µm thick. Hymenial surface smooth to have small verrucous process, pruinose, white when fresh, white to slightly cream on drying; lightly darkening in KOH. Margin sterile, narrow, white, attached.

**Hyphal system**—Hyphal system monomitic, generative hyphae simple-septa, colorless, thick-walled, unbranched, interwoven, 3–4.5 µm in diameter, subhymenial hyphae densely covered by larger crystals, basal hyphae regular; IKI–, CB–; tissues unchanged in KOH.

**Hymenium**—Hymenial cystidia and cystidoles absent; basidia clavate to subcylindrical, with four sterigmata and a simple-septum, 13–24 µm × 3.5–4.5 µm.

**Spores**—Basidiospores cylindrical, colorless, thin-walled, smooth, IKI–, CB–, (3.3–) 3.5–6.7(–7) µm × 1.5–2.7(–2.9) µm, L = 4.42 µm, W = 1.94 µm, Q = 2.21–2.35 (n = 60/2).

**Additional specimen examined**—China, Yunnan Province, Zixishan National Forestry Park, on fallen branch of angiosperm, 101.4° E, 25.1° N, 1 July 2018, C.L. Zhao 7112 (SWFC).

**Habitat and ecology**—Climate of the sample collection site is monsoon humid, and the forest type is evergreen broad-leaved forest, and the samples were collected on an angiosperm branch.

***Phanerochaete rhizomorpha*** C.L. Zhao and D.Q. Wang sp. nov. Figure 5 and Figure 6.

MycoBank no.: MB 841272.

Diagnosis: It differs from *P. citrinosanguinea* by its orange hymenial surface and larger cystidia 48.5–71.5 µm × 3–6.5 µm)

**Holotype**—China, Yunnan Province, Dali, Nanjian Country, Lingbaoshan National Forestry Park, on the fallen branch of angiosperm, 24.7° N, 100.6° E, 10 January 2019, C.L. Zhao 10,477 (SWFC).

**Etymology**—***rhizomorpha*** (Lat.): from Latin, referring to the rhizomorphic basidiomata of the specimens.

**Fruiting body**—Basidiomata annual, resupinate, adnate, easily detachable from substrate, membranaceous, up to 5 cm long, 3 cm wide, 200–300 µm thick. Hymenial surface smooth, slightly orange when fresh, orange upon drying; lightly darkening in KOH. Margin sterile, buff to slightly orange, up to 1 mm wide, rhizomorphic.

**Hyphal system**—Hyphal system monomitic, generative hyphae simple-septa, colorless, thick-walled, frequently branched, interwoven, 3–6.5 µm in diameter, basal hyphae regular, numerous crystals present among the abhymenium hyphae, IKI–, CB–; tissues unchanged in KOH.

**Hymenium**—Hymenium cystidia subulate or tapering, colorless, thick-walled, with 2–4 septa, 48.5–71.5 µm × 3–6.5 µm; basidia subcylindrical, with 4 sterigmata, 18.5–35.5 µm × 3.5–5.5 µm.

**Spores**—Basidiospores narrower ellipsoid to ellipsoid, colorless, thin-walled, smooth, with oil 1–2 drops inside, IKI–, CB–, 4.5–5.8(–6) µm × 2.7–3.6(–3.8) µm, L = 5.07 µm, W = 3.19 µm, Q = 1.58–1.60 (n = 62/2).

**Additional specimen examined**—China, Yunnan Province, Nanjian Country, Lingbaoshan National Forestry Park, on fallen branch of angiosperm, 24.7° N, 100.6° E, 10 January 2019, C.L. Zhao 10,470 (SWFC).

**Habitat and ecology**—Climate of the sample collection site is a transition between tropical and subtropical climate, and the forest type is the tropical monsoon evergreen broad-leaved forest, and the samples were collected on an angiosperm trunk.

## 4. Discussion

In the present study, two new species, *Phanerochaete pruinosa* C.L. Zhao and D.Q. Wang and *P*. *rhizomorpha* C.L. Zhao and D.Q. Wang spp. nov., are described based on phylogenetic analyses and morphological characters. The nucleotide differences of phylogenetically similar species to *Phanerochaete pruinosa* and *P*. *rhizomorpha*.

Phylogenetically, Xu et al. [19] revealed the taxonomy and phylogeny of *Phanerochaete sensu stricto* (Polyporales, Basidiomycota) with emphasis on Chinese collections, which showed that twenty-eight species of *Phanerochaete* s.s. from China are confirmed by morphology and DNA sequence data. In the present study (Figure 2), two new taxa clustered into *Phanerochaete* s.s., in which *P. pruinosa* was sister to *P. yunnanensis*, and then grouped with *P. robusta*. Another species *P. rhizomorpha* was closely grouped with *P. citrinosanguinea* with lower supports, and then grouped with *P. pseudosanguinea* and *P. sanguinea*. However, morphologically, *P. yunnanensis* is separated from *P. pruinosa* by having a pale orange to greyish orange and densely cracked hymenial surface [19]; *P. robusta* differs in its yellow basidiomata and two kinds of cystidia without encrustation, larger basidiospores (5.5–7 μm × 2.4–2.9 μm) and a boreal distribution [34]. *Phanerochaete citrinosanguinea* differs from *P. rhizomorpha* by having yellow to reddish yellow hymenial surface, and smaller cystidia (31–48 μm × 2.3–4.8 µm) [21]; *P. pseudosanguinea* differs *P. rhizomorpha* in its light red or dark red hymenial surface, and narrower basidiospores (4–5.5 µm × 2–2.5 µm) [21]; *P.*
*sanguinea* is separated from *P. rhizomorpha* by having the thin-walled cystidia and the larger basidia (25–45 μm × 4–6 μm) [1]; in addition, there is some coloration of wood as in *P. sanguinea*.

Morphologically, *Phanerochaete pruinosa* is similar to *P. concrescens* Spirin and Volobuev and *P. sordida* (P. Karst.) J. Erikss. and Ryvarden, based on presence of white or cream hymenial surface. However, *P. concrescens* differs from *P. pruinosa* by having the large basidia (27–39 μm × 4–5 µm) [34]; *P. sordida* is separated from *P. pruinosa* by presence of cystidia and wider basidiospores (5–7 μm × 2.5–3.5 μm) [1].

*Phanerochaete rhizomorpha* reminds four taxa of *Phanerochaete* based on the character by having the rhizomorph, *P. burdsallii* Y.L. Xu, Nakasone and S.H. He, *P. leptocystidiata* Y.L. Xu and S.H. He, *P. sinensis* Y.L. Xu, C.C. Chen and S.H. He and *P. subrosea* Y.L. Xu and S.H. He. However, *P*. *burdsallii* differs from *P.*
*rhizomorpha* by having the cystidia encrusted with small crystals [19]; *P.*
*leptocystidiata* differs in having a tuberculate hymenial surface and thin-walled cystidia encrusted at apex (24–30 μm × 4–6 µm) [19]; *P. sinensis* differs in its thin-walled cystidia and the shorter basidia (17–22 μm × 4–5 µm) [19]; *P. subrosea* is separated from *P. rhizomorpha* by having the thin-walled and smaller cystidia (33–55 μm × 3–5 µm) [19].

*Phanerochaete rhizomorpha* is similar to *P. aurantiobadia* Ghob.-Nejh., S.L. Liu, Langer and Y.C. Dai, *P. cumulodentata* (Nikol.) Parmasto and *P. hymenochaetoides* Y.L. Xu and S.H. He based on the character by the orange hymenial surface. However, *P. aurantiobadia* differs from *P. rhizomorpha* by having the larger basidiospores (5–8.3 μm × 2–3 µm) [16]; *P. cumulodentata* differs from *P. rhizomorpha* by a tuberculate hymenophore and shorter basidia (16.7–28.3 μm × 3.7–5.2 µm) [34]; *P. hymenochaetoides* differs from *P. rhizomorpha* by having both smaller cystidia (30–45 μm × 3–4 µm) and basidiospores (4–5.2 μm × 2–2.8 µm) [19].

In the ecology and biogeography, the taxa of *Phanerochaete* are a typical example of wood-rotting fungi, which are mainly distributed in Asia, Europe, and America, and the substrata are mostly hardwood [1,25], and this genus is an extensively studied group of Basidiomycota; nonetheless, the wood-rotting fungi diversity is still not well known in the subtropics and tropics [44,45,46,47,48]. The two new species, *Phanerochaete pruinosa* and *P*. *rhizomorpha* spp. nov., were found in subtropics, which enriches the diversity of wood-rotting fungi.

## Figures and Tables

**Figure 1 jof-07-01063-f001:**
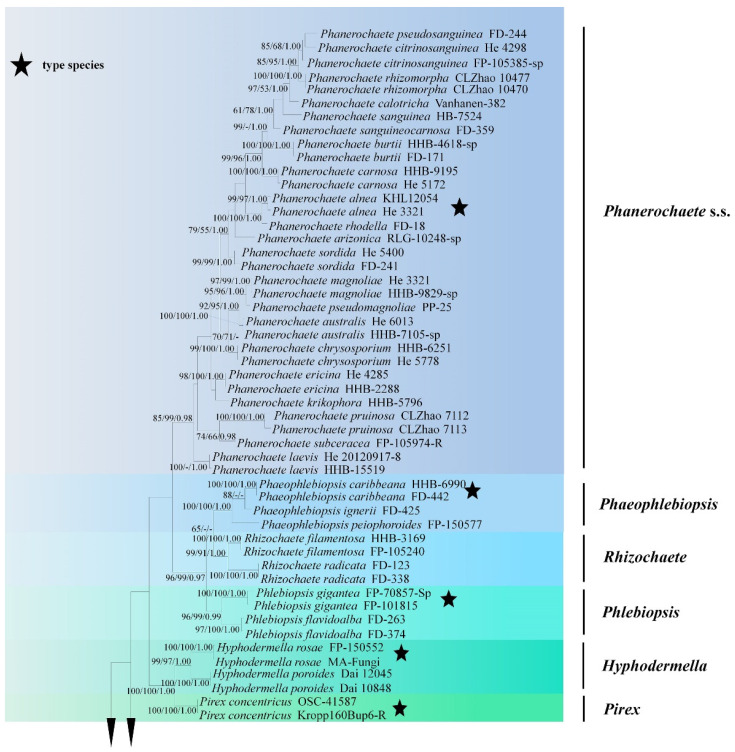
Maximum Parsimony strict consensus tree illustrating the phylogeny of two new species and related genera in *Phanerochaetaceae* and Irpicaceae based on ITS+nLSU sequences. Branches are labeled with maximum likelihood bootstrap values >70%, parsimony bootstrap values >70% and Bayesian posterior probabilities >0.95, respectively.

**Figure 2 jof-07-01063-f002:**
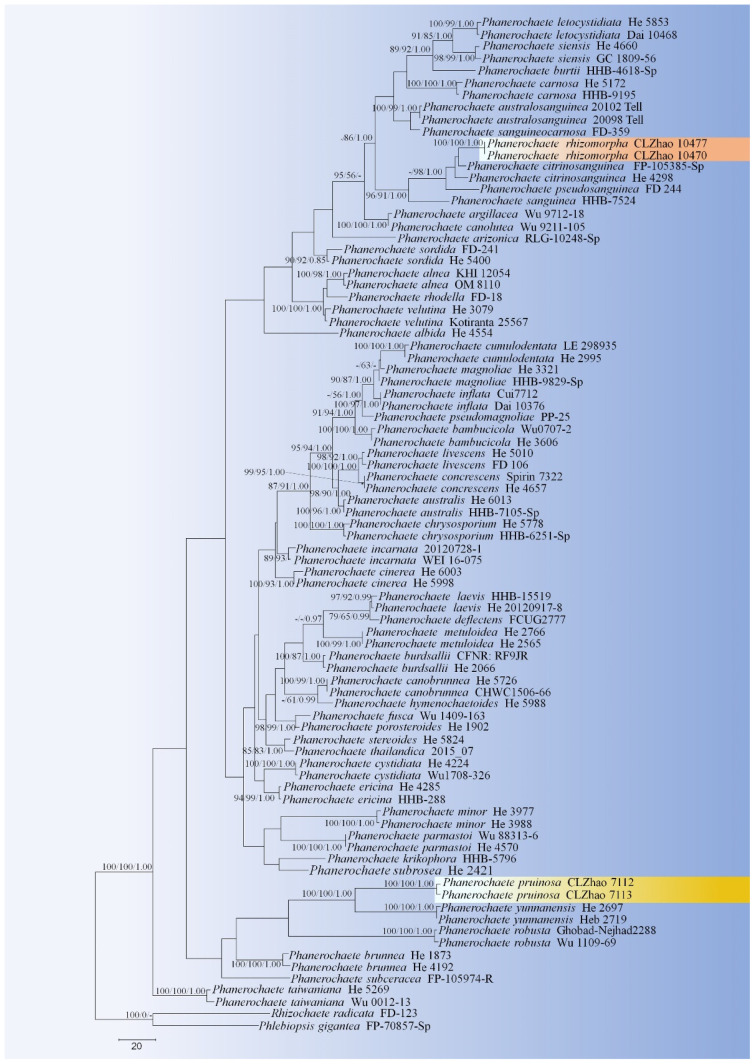
Maximum Parsimony strict consensus tree illustrating the phylogeny of two new species and related species in *Phanerochaete* based on ITS+nLSU sequences. Branches are labeled with maximum likelihood bootstrap values >70%, parsimony bootstrap values >70% and Bayesian posterior probabilities >0.95, respectively. The yellow backgrounds indicate new species.

**Figure 3 jof-07-01063-f003:**
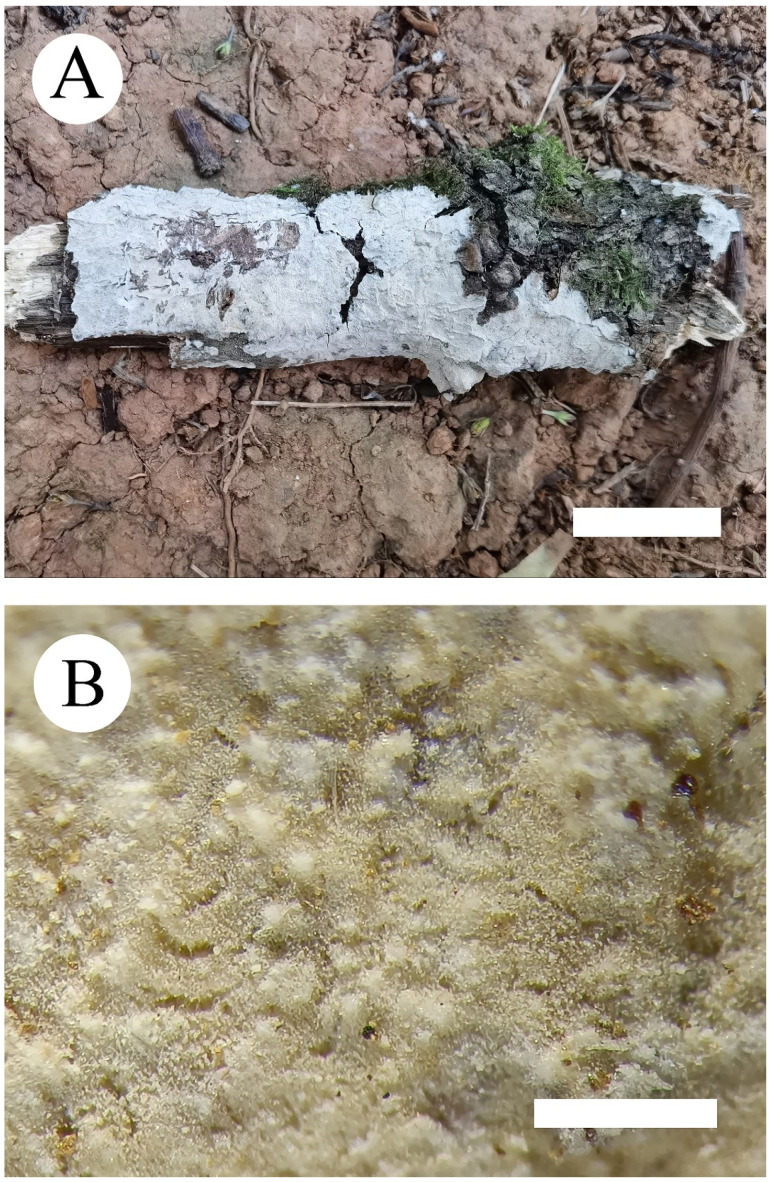
Basidiomata of *Phanerochaete pruinosa* (holotype) Bars: (**A**) = 2 cm and (**B**) = 1 mm.

**Figure 4 jof-07-01063-f004:**
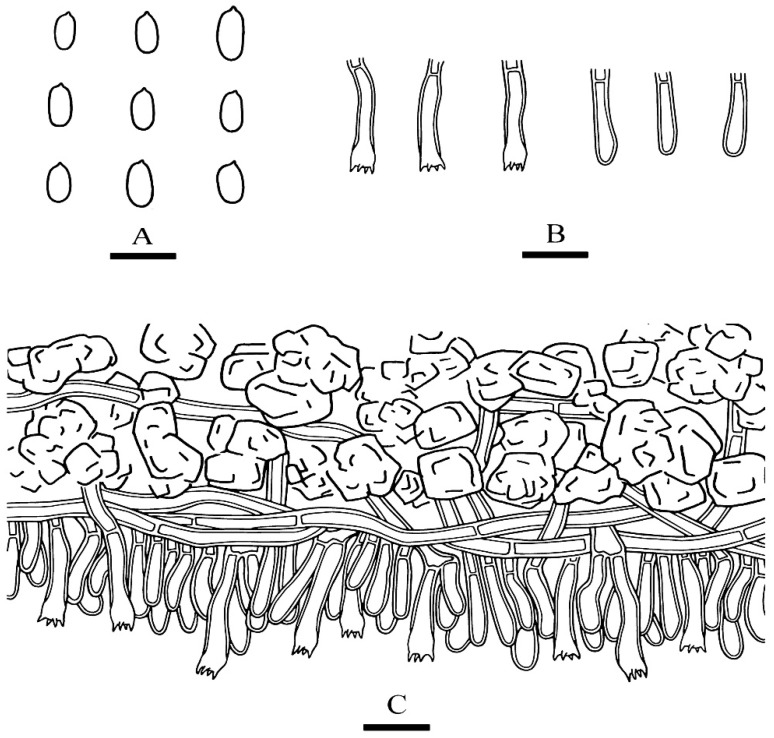
Microscopic structures of *Phanerochaete pruinosa* (holotype): basidiospores (**A**), basidia and basidioles (**B**), A section of hymenium (**C**). Bars: (**A**) = 5 µm, (**B**,**C**) = 10 µm.

**Figure 5 jof-07-01063-f005:**
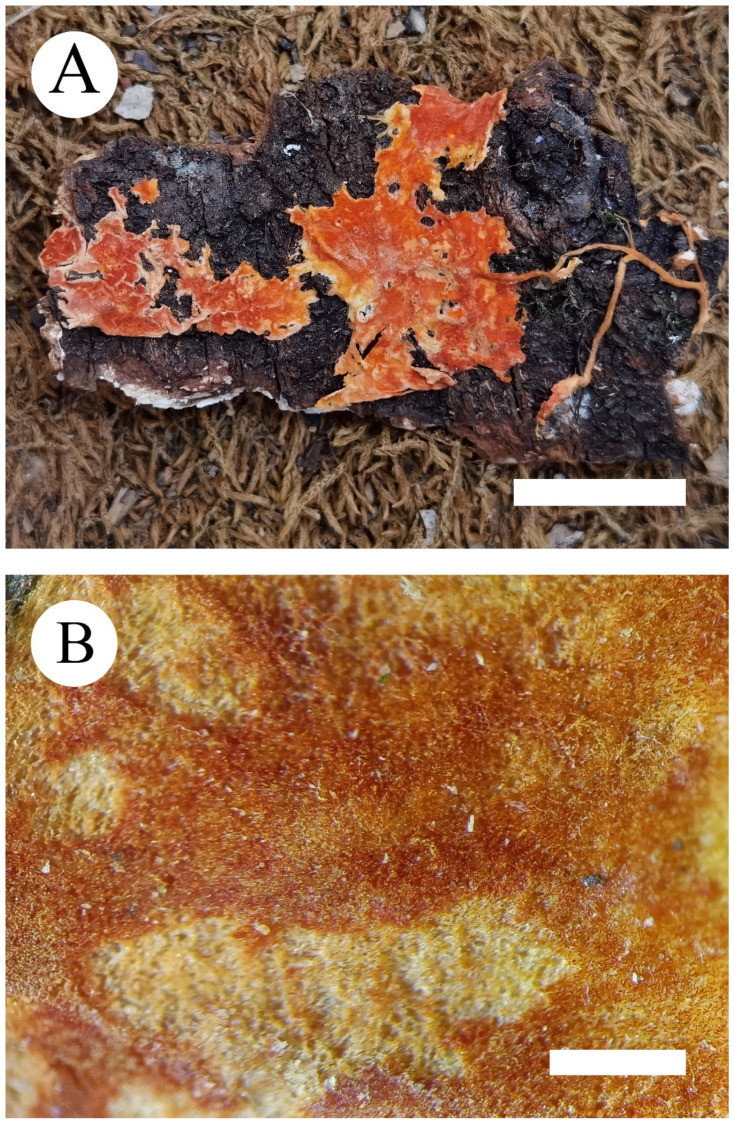
Basidiomata of *Phanerochaete rhizomorpha* (holotype) Bars: (**A**) = 2 cm and (**B**) = 1 mm.

**Figure 6 jof-07-01063-f006:**
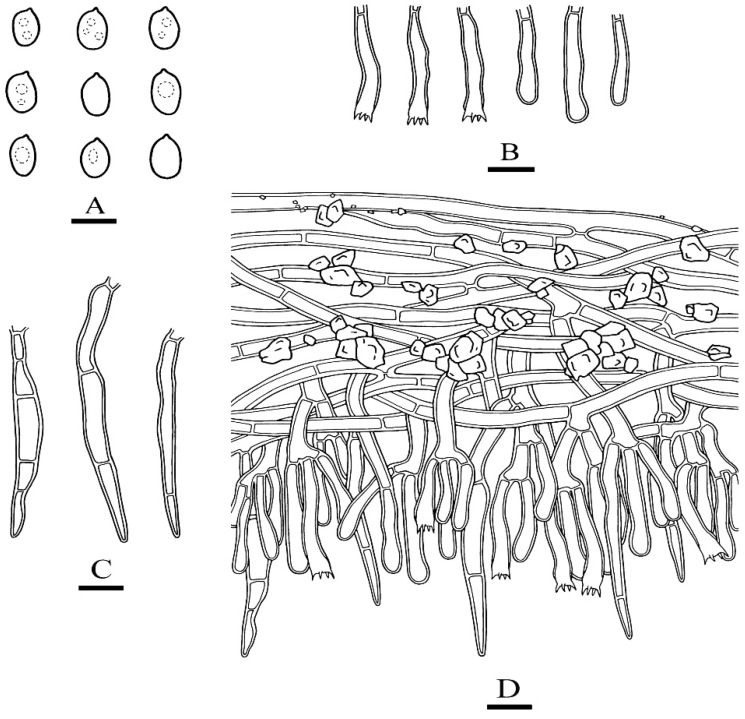
Microscopic structures of *Phanerochaete rhizomorpha* (holotype): basidiospores (**A**), basidia and basidioles (**B**), cystidia (**C**). A section of hymenium (**D**). Bars: (**A**) = 5 µm, (**B**–**D**) = 10 µm.

**Table 1 jof-07-01063-t001:** List of species, specimens, and GenBank accession numbers of sequences used in this study.

Species Name	Specimen No.	GenBank Accession No.		References
		ITS	nLSU	
*Bjerkandera adusta*	FP-101236	KP134982		[21]
*B. adusta*	HHB-12826	KP134983	KP135198	[21]
*B. fumosa*	Dai 12674B	MW507112	MW520213	[28]
*B. fumosa*	Dai 21087	MW507110		[28]
*Byssomerulius corium*	FP-102382	KP135007	KP135230	[21]
*B. corium*	FP-107055	KP135008		[21]
*Ceraceomyces serpens*	HHB-15692-Sp	KP135031	KP135200	[21]
*C. serpens*	L-11105	KP135032		[21]
*Ceriporia purpurea*	KKN-223-Sp	KP135044	KP135203	[21]
*C. purpurea*	HHB-3964	KP135042		[21]
*C. reticulata*	RLG-11354	KP135041	KP135204	[21]
*C. reticulata*	L-7837	KP135040		[21]
*Efibula gracilis*	FD-455	KP135027	MZ637116	[21]
*E. gracilis*	FP-102052	KP135028		[21]
*E. tropica*	Wu 0809-8	MZ636968	MZ637130	unpublished
*E. tropica*	WEI 18-149	MZ636967	MZ637129	unpublished
*Gloeoporus dichrous*	FP-151129	KP135058	KP135213	[21]
*G. pannocinctus*	L-15726-Sp	KP135060	KP135214	[21]
*Hyphodermella poroides*	Dai 12045	KX008367	KX011852	[29]
*H. poroides*	Dai 10848	KX008368	KX011853	[29]
*H. rosae*	FP-150552	KP134978	KP135223	[21]
*H. rosae*	MA-Fungi	FN600389	JN939588	[30]
*Irpex lacteus*	FD-9	KP135026	KP135224	[21]
*I. lacteus*	FD-93	KP135025		[21]
*Meruliopsis albostramineus*	HHB-10729	KP135051	KP135229	[21]
*M. albostramineus*	L-9778	KP135052		[21]
*M. taxicola*	CBS 45548	MH856432	MH867978	[31]
*M. taxicola*	Kuljok 00/75 (GB)	EU118648		[32]
*Phaeophlebiopsis caribbeana*	HHB-6990	KP135415	KP135243	[21]
*P. caribbeana*	FD-442 (TYPE)	KP135416		[21]
*P. ignerii*	FD-425	KP135418		[21]
*P. peiophoroides*	FP-150577	KP135417	KP135273	[21]
*Phanerochaete. albida*	FD-31	KP135308	KP135210	[19]
*P. alnea*	OM 8110	KP135171		[21]
*P. alnea*	KHL 12054	EU118653	EU118653	[32]
*P. argillacea*	Wu 9712-18		GQ470656	[13]
*P. arizonica*	RLG-10248-sp	KP135170	KP135239	[21]
*P. australis*	He 6013	MT235656	MT248136	[19]
*P. australis*	HHB-7105-sp	KP135081	KP135240	[21]
*P. australosanguinea*	20098 Tell		MH233928	[33]
*P. australosanguinea*	20102 Tell		MH233929	[33]
*P. bambucicola*	He 3606	MT235657	MT248137	[19]
*P. bambucicola*	Wu 0707-2	MF399404	MF399395	[15]
*P. brunnea*	He 4192	MT235658	MT248138	[19]
*P. brunnea*	He 1873	KX212220	KX212224	[17]
*P. burdsallii*	He 2066	MT235690	MT248177	[19]
*P. burdsallii*	CFMR: RF9JR	KU668973		unpublished
*P. burtii*	HHB-4618-sp	KP135117	KP135241	[21]
*P. burtii*	FD-171	KP135116		[21]
*P. calotricha*	Vanhanen-382	KP135107		[21]
*P. canobrunnea*	He 5726	MT235659	MT248139	[19]
*P. canobrunnea*	CHWC 1506-66	LC412095	LC412104	[14]
*P. canolutea*	Wu 9211-105		GQ470641	[13]
*P. carnosa*	He 5172	MT235660	MT248140	[19]
*P. carnosa*	HHB-9195	KP135129	KP135242	[21]
*P. chrysosporium*	HHB-6251	KP135094	KP135246	[21]
*P. chrysosporium*	He 5778	MT235661	MT248141	[19]
*P. cinerea*	He 6003		MT248172	[19]
*P. citrinosanguinea*	He 4298	MT235691	MT248178	[19]
*P. citrinosanguinea*	FP-105385-sp	KP135100	KP135234	[21]
*P. concrescens*	He 4657	MT235662	MT248142	[19]
*P. concrescens*	Spirin 7322	KP994380	KP994382	[34]
*P. cumulodentata*	He 2995	MT235664	MT248144	[19]
*P. cumulodentata*	LE 298935	KP994359	KP994386	[34]
*P. cystidiata*	He 4224	MT235665	MT248145	[19]
*P. cystidiata*	Wu 1708-326	LC412097	LC412100	[14]
*P. deflectens*	FCUG 2777		GQ470644	[13]
*P. ericina*	He 4285	MT235666	MT248146	[19]
*P. ericina*	HHB-2288	KP135167	KP135247	[21]
*P. exilis*	HHB-6988	KP135001	KP135236	[21]
*P. fusca*	Wu 1409-163	LC412099	LC412106	[14]
*P. hymenochaetoides*	He 5988		MT248173	[19]
*P. incarnata*	He 20120728-1	MT235669	MT248149	[19]
*P. incarnata*	WEI 16-075	MF399406	MF399397	[15]
*P. inflata*	Dai 10376	JX623929	JX644062	[35]
*P. inflata*	Cui 7712	JX623930	JX644063	[35]
*P. krikophora*	HHB-5796	KP135164	KP135268	[21]
*P. laevis*	He 20120917-8	MT235670	MT248150	[19]
*P. laevis*	HHB-15519	KP135149	KP135249	[21]
*P. leptocystidiata*	He 5853	MT235685	MT248168	[19]
*P. leptocystidiata*	Dai 10468	MT235684	MT248167	[19]
*P. livescens*	He 5010	MT235671	MT248151	[19]
*P. livescens*	FD-106	KP135070	KP135253	[21]
*P. magnoliae*	He 3321	MT235672	MT248152	[19]
*P. magnoliae*	HHB-9829-sp	KP135089	KP135237	[21]
*P. metuloidea*	He 2565		MT248163	[19]
*P. metuloidea*	He 2766	MT235682	MT248164	[19]
*P. minor*	He 3988	MT235686	MT248170	[19]
*P. minor*	He 3977		MT248169	[19]
*P. parmastoi*	He 4570	MT235673	MT248153	[19]
*P. parmastoi*	Wu 880313-6		GQ470654	[13]
*P. porostereoides*	He 1902	KX212217	KX212221	[17]
** *P. pruinosa* **	**CLZhao 7712**	**MZ435346**	**MZ435350**	**Present study**
** *P. pruinosa* **	**CLZhao 7713**	**MZ435347**	**MZ435351**	**Present study**
*P. pseudomagnoliae*	PP-25	KP135091	KP135250	[21]
*P. pseudosanguinea*	FD-244	KP135098	KP135251	[21]
*P. queletii*	HHB-11463	KP134994	KP135235	[21]
*P. queletii*	FP-102166	KP134995		[21]
** *P. rhizomorpha* **	**CLZhao 10470**	**MZ435348**	**MZ435352**	**Present study**
** *P. rhizomorpha* **	**CLZhao 10477**	**MZ435349**	**MZ435353**	**Present study**
*P. rhodella*	FD-18	KP135187	KP135258	[21]
*P. robusta*	Wu 1109-69	MF399409	MF399400	[15]
*P. robusta*	Ghobad 2288	KP127068	KP127069	[16]
*P. sanguinea*	HHB-7524	KP135101	KP135244	[21]
*P. sanguineocarnosa*	FD-359	KP135122	KP135245	[21]
*P. sinensis*	He 4660	MT235688	MT248175	[19]
*P. sinensis*	GC 1809-56	MT235689	MT248176	[19]
*P. sordida*	He 5400	MT235676	MT248157	[19]
*P. sordida*	FD-241	KP135136	KP135252	[21]
*P. stereoides*	He 5824	MT235677	MT248158	[19]
*P. subceracea*	FP-105974-R	KP135162	KP135255	[21]
*P. subrosea*	He 2421	MT235687	MT248174	[19]
*P. taiwaniana*	He 5269	MT235680	MT248161	[19]
*P. taiwaniana*	Wu 0112-13	MF399412	MF399403	[15]
*P. thailandica*	2015_07	MF467737		[36]
*P. velutina*	He 3079	MT235681	MT248162	[19]
*P. velutina*	Kotiranta 25567	KP994354	KP994387	[34]
*P. xerophila*	HHB-8509-Sp	KP134996	KP135259	[21]
*P. xerophila*	KKN-172	KP134997		[21]
*P. yunnanensis*	He 2719	MT235683	MT248166	[19]
*P. yunnanensis*	He 2697		MT248165	[19]
*Phlebiopsis flavidoalba*	FD-263	KP135402	KP135271	[21]
*P. flavidoalba*	FD-374	KP135403		[21]
*P. gigantea*	FP-70857-sp	KP135390	KP135272	[21]
*P. gigantea*	FP-101815	KP135389		[21]
*Pirex concentricus*	OSC-41587	KP134984	KP135275	[21]
*P. concentricus*	Kropp160Bup6-R	KP134985		[21]
*Rhizochaete filamentosa*	HHB-3169	KP135410	KP135278	[21]
*R. filamentosa*	FP-105240	KP135411		[21]
*R.radicata*	FD-123	KP135407	KP135279	[21]
*Terana caerulea*	FP-104073	KP134980	KP135276	[21]
*T. caerulea*	T-616	KP134981		[21]
*Trametopsis aborigena*	Robledo 1238	KY655337		[37]
*T. aborigena*	Robledo 1236	KY655336		[37]
*T. cervina*	AJ-185	JN165020	JN164839	[21]
*T. cervina*	AJ-189	JN165021		[21]

New sequences are shown in bold.

## Data Availability

Publicly available datasets were analyzed in this study. This data can be found here: [https://www.ncbi.nlm.nih.gov/, https://www.mycobank.org/; https://www.treebase.org/treebase-web/home.html;jsessionid=6440D6056D96C04A8D29290992C18900, submission ID 28442; accessed on 16 November 2021].

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
