# Peer review of "Morphological and Phylogenetic Evidence for Recognition of Two New Species of Phanerochaete from East Asia"

_jof, 2021, doi:10.3390/jof7121063_

Round 1

Reviewer 1 Report

This manuscript describes two new species of Phanerochaete using morphology as well as phylogeny of nuclear ribosomal ITS and LSU regions, which are popular and standard molecular regions in basidiomycete phylogenetic studies. The two species are probably new, as they apparently show significant nucleotide/morphological differences with similar-looking species.

However, the manus needs some substantial improvements before it can be further considered for publication. The main issue concerns the generic assignment of the new species and the vague taxon sampling for the phylogeny. Phanerochaete is a large and highly polyphyletic genus with several satellite genera. While the authors claim that their two new species belong to Phanerochaete s.s., however, they do not provide any information on its definition and delimitation. What is actually Phanerochaete s.s., and how would the authors prove that their new species belong to it? The phylogenetic tree in Fig. 1 is neatly composed of a large number of Phanerochaete species, with only two species from other genera assigned as outgroup. The tree does not reflect the heavily polyphyletic nature of Phanerochaete and its various clades. It is not clear which clade represents the Phanerochaete s.s., and whether or not the new species are closely related to it or not. Both new species show considerable distance to the generic type P. alnea. So the authors must deeply think about reconstructing the phylogeny to clearly show the clade representing the core Phanerochaete, and the probably competing clades/genera.

Other issues are briefly noted in the following:

  • Title: The first phrase (The hidden corticioid fungal diversity) is absolutely redundant, as it does not match the content of the text.
  • The abstract mentions allantoid spores for P. pruinosa, while in the description and Fig. 3 they are otherwise. Please correct.
  • The last two sentences in the abstract are not much informative. They must be replaced with main diagnostic characters of the new species compared to the most resembling species.
  • The cystidia in P. rhizomorpha are subulate and not cylindrical. Please correct.
  • In the Material and Methods, the taxon sampling strategy must be clearly explained to show how, and based on which criteria, the taxa were selected for phylogeny, with no intentional bias.
  • In the Taxonomy section, both new species need a short Diagnosis (most important differencing characters towards resembling species).
  • Please correct the etymology for pruinosa, as there is basically no “white powder on hymenial surface”!
  • Delete Table 2, it is a copy of blast searches and not suitable for publication.
  • Discussion: The last paragraph does not make sense and its statements on worldwide distribution of Phanerochaete are not substantiated. Please delete it. Accordingly, Fig. 6 is completely redundant and uninformative.
  • The language of the text is poor or not clear in several parts. A thorough language revision is needed.

Author Response

Dear editor,
I am so sorry to bother you again.
We are very grateful to you for your patient comments on our manuscript. We have
carefully revised the manuscript directly in the text, according to these comments.
The responses to the comments were listed below and highlighted in green color.
Could you please contact me if you have any questions or concerns ?
Thanks for your kindness.
Warm regards,
Dong-Qiong Wang & Chang-Lin Zhao
---------------------------------------------
Reviewer's comments:
Reviewer:
Comments in general:
This manuscript describes two new species of Phanerochaete using morphology as well as phylogeny of nuclear ribosomal ITS and LSU regions, which are popular and standard molecular regions in basidiomycete phylogenetic studies. The two species are probably new, as they apparently show significant nucleotide/morphological
differences with similar-looking species.

However, the manus needs some substantial improvements before it can be further considered for publication. The main issue concerns the generic assignment of the new species and the vague taxon sampling for the phylogeny. Phanerochaete is a large and highly polyphyletic genus with several satellite genera. While the authors claim that
their two new species belong to Phanerochaete s.s., however, they do not provide any information on its definition and delimitation. What is actually Phanerochaete s.s., and how would the authors prove that their new species belong to it? The phylogenetic tree in Fig. 1 is neatly composed of a large number of Phanerochaete species, with only two species from other genera assigned as outgroup. The tree does not reflect the heavily polyphyletic nature of Phanerochaete and its various clades. It is not clear which clade represents the Phanerochaete s.s., and whether or not the new species are closely related to it or not. Both new species show considerable distance to the generic type P. alnea. So the authors must deeply think about reconstructing the phylogeny to clearly show the clade representing the core Phanerochaete, and the probably competing clades/genera

Response: We have added a new phylogenetic tree (Fig. 1) within a larger scale in the family Phanerochaetaceae according to the reviewer’s comment.

Comments in text:
Title
1) The first phrase (The hidden corticioid fungal diversity) is absolutely redundant, as it does not match the content of the text.;
Response: We have deleted it according to the reviewer’s comment. Abstract
1) The abstract mentions allantoid spores for P. pruinosa, while in the description and Fig. 3 they are otherwise. Please correct.;
Response: We checked and modified it to be nearly cylindrical.
2) The last two sentences in the abstract are not much informative. They must be replaced with main diagnostic characters of the new species compared to the most resembling species.;
Response: We have revised it as “These phylogenetic showed that the two new species clustered into Phanerochaete, in which P. pruinosa was sister to P. yunnanensis with high supports (100% BS, 100% BT, 1.00 BPP); morphologically differing by a pale orange to greyish orange and densely cracked hymenial surface. Another species P. rhizomorpha was closely grouped with P. citrinosanguinea with lower supports; morphologically having yellow to reddish yellow hymenial surface, and smaller cystidia (31–48 μm × 2.3–4.8 µm).” according to the reviewer’s comment.

Materials and methods
1) In the Material and Methods, the taxon sampling strategy must be clearly explained to show how, and based on which criteria, the taxa were selected for phylogeny, with no intentional bias.;
Response: We have added it as “The fruiting bodies were observed growing on the ground of broad-leaved treemixed forest. The fruiting bodies were dried in an electric food dehydrator at 40 °C, then sealed and stored in an envelope bag. They were then transported to Kunming where microscopic morphology and phylogeny would be studied at the mycology laboratory of Southwest Forestry University”.

Taxonomy
1) The cystidia in P. rhizomorpha are subulate and not cylindrical. Please correct.;
Response: We have revised it according to the reviewer’s comment.
2) In the Taxonomy section, both new species need a short Diagnosis (most important differencing characters towards resembling species).;
Response: We have added a short Diagnosis for both new taxa.
3) Please correct the etymology for pruinosa, as there is basically no “white powder on hymenial surface”!;
Response: We have added the new picture to show the powder.
4) Delete Table 2, it is a copy of blast searches and not suitable for publication.;
Response: We have deleted it according to the reviewer’s comment.

Discussion
1) The last paragraph does not make sense and its statements on worldwide distribution of Phanerochaete are not substantiated. Please delete it. Accordingly, Fig. 6 is completely redundant and uninformative.;
Response: We have deleted it according to the reviewer’s comment.
2) The language of the text is poor or not clear in several parts. A thorough language revision is needed;
Response: We have revised them according to the reviewer’s comment.

Reviewer 2 Report

Dear Authors,

generally, improve your descriptions of fruitbodies and the order of features should be arranged in the standard way as is possible find in monographic works about corticioid fungi.

What about hymenophore?, smooth? Add some features of margin fruitbodies (attached?, fimbriate?). What some changes of surface fruitbodies with KOH solution? It is important in some species of Phanerochaete. What about basal hyphae? What about biotope of both described species?

Please checked English carefully.

You find some remarks in attached file. There are lines and some remarks to them, moreover proposed changes are shown in bold.

Author Response

Dear editor,

I am so sorry to bother you again.

We are very grateful to you for your patient comments on our manuscript. We have carefully revised the manuscript directly in the text, according to these comments.

The responses to the comments were listed below and highlighted in green color.

Could you please contact me if you have any questions or concerns.

Thanks for your kindness.

Warm regards,

Dong-Qiong Wang & Chang-Lin Zhao

---------------------------------------------

Reviewer's comments:

Reviewer:

Comments in general:

1) generally, improve your descriptions of fruitbodies and the order of features should be arranged in the standard way as is possible find in monographic works about corticioid fungi.

Response: We have checked and added some descriptions.

2) What about hymenophore? smooth?

Response: We have checked it carefully and it is smooth hymenophore.

3) Add some features of margin fruitbodies (attached?, fimbriate?).

Response: We have added some features of margin fruitbodies according to the reviewer’s comment.

4) What some changes of surface fruitbodies with KOH solution? It is important in some species of Phanerochaete.

Response: We have carried out tests and found that both fruitbodies are significantly blackened under KOH.

5) What about basal hyphae?

Response: We have revised them according to the reviewer’s comment.

6) What about biotope of both described species?

Response: We've added the biotope for two species as “Habitat and ecology—Climate of the sample collection site is a transition between tropical and subtropical climate, and the forest type is the tropical monsoon evergreen broad-leavedforest, and the samples were collected on an angiosperm trunk”.

Comments in text:

Abstract

1) 11. “Corticioid fungi is a large group of Basidiomycota”, – unnecessary in abstract;

Response: We have deleted it.

2) 15-16. “the smooth hymenophore covering orange hymenial surface”, – Used indefinite articles and revised “covering” as “covered by”;

Response: We have revised it according to the reviewer’s comment.

3) 18. Revised “rhizomorph” as “rhizomorphs”;

Response: We have revised it.

4) 19. Added “analyses”;

Response: We have revised it according to the reviewer’s comment.

Introduction

1). 29.Deleted “and mushrooms”;

Response: We have deleted it.

2) 33. Added “of”;

Response: We have revised it according to the reviewer’s comment.

3) 33-34. Deleted “in which P. chrysosporium Burds. nested into phlebioid clade in Polyporales and grouped with Bjerkandera adusta (Willd.) P. Karst.”;

Response: We have deleted them.

4) 43-44. Deleted “(Polyporales, Basidiomycota)”;

Response: We have revised it according to the reviewer’s comment.

5) 45-45. “…across nine lineages in the phlebioid clade, in which 6 lineages”, – Verbally, as a nine in the same line;

Response: We have revised it as “six” according to the reviewer’s comment.

6) 47. “…….described genera [21]”, – Added “.” after “[21]”;

Response: We have added it.

7) 47-50. “the macromorphology of fruiting bodies and hymenophore construction did not reflect monophyletic groups”, – Exact formulation from the cited article;

Response: We have revised it as “Miettinen et al. [22] explored the DNA-phylogeny-based and morphology-based to reconcile the Polypores and genus concepts in the family Phanerochaetaceae, that the macromorphology of fruiting bodies and hymenophore construction did not reflect monophyletic groups,” according to the reviewer’s comment.

8) 53-56. “Recently, the research supported by morphological studies and the phylogenetic analyses, showed that many new taxa of Phanerochaete s.s. were found and displayed the taxonomic status for the new taxa within Phanerochaete s.s. [14,19].”, – using “Phanerochaete s.s.” twice seems illogical.;

Response: We have revised it according to the reviewer’s comment.

9) 59. Revised “evidences” as “evidence”;

Response: We have revised it.

10) 61. Added “and”;

Response: We have added it according to the reviewer’s comment.

Materials and methods

1) 78. “(b)”, – Move at the end of sentence.;

Response: We have revised it according to the reviewer’s comment.

2) 93. Table 1. – What does it mean asterisk in some species? ;

Response: We have deleted it.

3) 100. “……Rhizochaete radicata Henn”, – It is necessity fill all authors;

Response: We added the names of all the authors.

4) 103-104. “Its approaches followed Zhao and Wu [34], and the tree construction procedure was performed in PAUP* version 4.0b10 [35]”, – I do not understand, 34 is for PAUP but 35 no.;

Response: We have revised them in whole text.

5) 107-108. Revised “[38]” as “[37]”; Revised “[39]” as “[38]”

Response: We have revised it according to the reviewer’s comment.

Taxonomy

1) 125-134. “In addition the results of BLAST queries in NCBI,……P. leptocystidiata (ident 99.13%, MT248167), P. sinensis (ident 99.42%, MT248175) in Table 2.” – Deleted this paragraph;

Response: We have revised it.

2) 157. Figure 2, – I am not sure with sufficient quality of resolution of phylogenetic tree;

Response: We have revised it according to the reviewer’s comment.

3) 157. Deleted Table 2;

Response: We have deleted it.

4) 163. Revised “and” as “&”;

Response: We have revised it according to the reviewer’s comment.

5) 168. “on fallen branch of angiosperm”, – on the bark or on the wood?

Response: We have revised “on fallen branch of angiosperm” as “on the bark of fallen angiosperms”.

6) 170. Figure 3, – “B” does not have sufficient quality in focus;

Response: We have revised the picture according to the reviewer’s comment.

7) 175. “Fruiting body—Basidiomata annual, resupinate, adnate, membranaceous”, – Confluent individual parts? Separable?

Response: It is confluent individual parts.

8) 182. “subhymenial hyphae”, – maybe rearrange some part of description, subhymnenial hyphae are not hymenium;

Response: We have revised it according to the reviewer’s comment.

9) 185. Revised “Hymenium” as “Hymenial”;

Response: We have revised it.

10) 185. “basidia clavate to subcylindrical”, – thickwalled as is seen on the figure?

Response: We have checked it carefully in the original pictures and it is thickwalled.

11) 186. Revised “4” as “four”;

Response: We have revised it according to the reviewer’s comment.

12) 191-192. “(A)、(B)、(C)” – Moved to the front of punctuation;

Response: We have checked and modified it.

13) 193. Revised “specimens” as “specimen”;

Response: We have revised it according to the reviewer’s comment.

14) 213-214. “…...up to 1 mm wide, rhizomorphic, hyphal cords present.” – Is there some difference between hyphal cords and rhizomorphs?

Response: We have deleted the hyphal cords, in which both are different.

15) 218-219. “Hymenium cystidia cylindrical, thick-walled, presence of 2–4 septa.”, – It seems as conical or tapering from the drawing?

Response: We have revised it as “Hymenium cystidia subulate or tapering”.

16) 219. Revised “presence of” as “with”;

Response: We have revised it according to the reviewer’s comment.

17) 221. “Spores” – not in italics;

Response: We have checked and modified it.

18) 221. “Basidiospores narrower ellipsoid to ellipsoid,” – narrowly?;

Response: We have revised it according to the reviewer’s comment.

19) 222. “with oil 1-2 drops inside” as “with 1–2 oil drops inside”;

Response: We have checked and modified it.

20) 196. “Figure 6” – what about stones in the fruitbody? There is any mention in the text;

Response: We have descripted as “numerous crystals present among the abhymenium hyphae”.

21) 225-226. “(A)、(B)、(C)” Moved to the front of punctuation;

Response: We have checked and modified it.

Discussion

1) 233. “Discussion” – Generally, there are more repeated sentences, text is too simple and unifom. Scientific names of newly mentioned Phanerochate need authors names!

Response: We have deleted the more repeated sentences and added the newly mentioned Phanerochate need authors names.

2) 234-236. “In the present study, two new species, Phanerochaete pruinosa and P. rhizomorpha spp. nov., are described based on phylogenetic analyses and orphological characters.” – It is same as 240-241.

Response:  We have revised it according to the reviewer’s comment.

3) 283. Deleted “Table 3”;

Response: We have deleted it according to the reviewer’s comment.

4) 242-243 “in which P. pruinosa was sister to P. yunnanensis with high supports, and then grouped with P. robusta without supported data”, – More or less exact repeated more times. It may be, at the first sight, that your species does not have molecular support. I recommend delete remarks about grouped to P. robusta;

Response: Response:  We have revised it.

5) 248. “…….and a boreal distribution [34];”, – Revised “;” as “.”;

Response: We have revised it according to the reviewer’s comment.

6) 248-249. “Phanerochaete citrinosanguinea differs from P. rhizomorpha by……”, – Start sentence with P. rhizomorpha, it is yours species;

Response: We have revised it according to the reviewer’s comment.

7) 250-251. “P. pseudosanguinea differs in its light red or dark red hymenial surface……” – It is too “far” from the above information on phylogeny. Maybe better solve at first P. pruinosa in this paragraph and then separately P. rhizomorpha?

Response: We have arranged it as phylogenetic part and taxonomy part.

8) 252-254. “P. sanguinea is separated from P. rhizomorpha by having the cracking hymenophore…….”, – It is not suitable feature, regularly there is any cracing in P. sanguinea. What about cystidia, thick-walled basidia as you draw? There is some coloration of wood as in P. sanguinea?

Response: We have revised it according to the reviewer’s comment.

9) 260-261. “Phanerochaete rhizomorpha reminds four species of Phanerochaete based on the character by having the hyphal cords.”, – Do you distinguish between hyphal cords and rhizomorphs? If you consider them as synonyms then more species are similar, at first P. sanguinea…May be mention genus Rhizochaete too.

Response: We have revised it as “rhizomorph”.

10) 268-271. Revised “cumulodentatum” as “cumulodentata”;

Response: We have revised it according to the reviewer’s comment.

11) 275-282. “In the ecology and biogeography……and many taxa were recently described from these areas [44–48].” – Too simple, mainly delete?, figure 7 is useful but the quality in resolution seems bad and there missing spreading some species in some countries, remarkable central-south-east Europe is without Phanerochate?;

Response: We have checked it and added all of Phanerochate species in Figure 6 as best as we can, according to the reviewer’s comment.

12) 293. Revised “were” as “was”;

Response: We have revised it.

13) 299-301. https://www.mycobank.org/page/Simple%20 277names%20search; http://purl.org/phylo/treebase, – I am not able connect to this exact internet link

Response: We have revised it.

Round 2

Reviewer 1 Report

The phylogeny has improved and the settlement of the two new species within Phanerochaete s.s. is fine.

However, some clarifications/corrections are needed in the text:

  • Taxon sampling: I can see that the taxa used in the phylogeny are apparently chosen mainly based on the study by Floudas & Hibbett 2015 (which is correct). If so, the authors should simply state this in their M&M, so that their taxon sampling strategy, (i.e. how taxa were selected for phylogeny) is clear to the readers. [Please note that the text you provided on lines 64-68 does not describe your “taxon sampling”, but it describes specimen collecting.]
  • Lines 279 onwards state that “… Phanerochaete are a typical example of wood-rotting fungi, which mainly distribute in Europe, such as …”. This is actually not correct. There are considerable number of Phanerochaete species described out of Europe. It is strange that the authors try to confer the global distribution of the genus by locating the phylogeny-sampled taxa in onto a map. As far as the presented phylogeny is just a representation of taxa and is not exhaustive, the distribution map out of it cannot be conclusive. So I do not believe the distribution map in Figure 7 has any use.
  • I strongly advise a thorough language revision for your text by an expert.

Author Response

Dear Editor,

I am so sorry to bother you again.

We are very grateful to you for your patient comments on our manuscript. We have carefully revised the manuscript directly in the text, according to these comments.

The responses to the comments were listed below and highlighted in green color.

Could you please contact me if you have any questions or concerns.

Thanks for your kindness.

Warm regards,

Dong-Qiong Wang & Chang-Lin Zhao

---------------------------------------------

Reviewer's comments:

Reviewer:

Comments in general:

1) Taxon sampling: I can see that the taxa used in the phylogeny are apparently chosen mainly based on the study by Floudas & Hibbett 2015 (which is correct). If so, the authors should simply state this in their M&M, so that their taxon sampling strategy, (i.e. how taxa were selected for phylogeny) is clear to the readers. [Please note that the text you provided on lines 64-68 does not describe your “taxon sampling”, but it describes specimen collecting.].

Response: We have added the sentences for it in the Material and Methods as “The taxon sampling strategy for the selection of sequences for phylogenetic trees was to choose 1) in a larger scale, focusing on the related genera in the families Phanerochaetaceae and Irpicaceae in Figure 1; 2) the related taxa based on BLAST search in GenBank within Phanerochaete s.l.; 3) all species of Phanerochaete s.s.”.

2) Lines 279 onwards state that “… Phanerochaete are a typical example of wood-rotting fungi, which mainly distribute in Europe, such as …”. This is actually not correct. There are considerable number of Phanerochaete species described out of Europe. It is strange that the authors try to confer the global distribution of the genus by locating the phylogeny-sampled taxa in onto a map. As far as the presented phylogeny is just a representation of taxa and is not exhaustive, the distribution map out of it cannot be conclusive. So I do not believe the distribution map in Figure 7 has any use.

Response: We have deleted this map according to the reviewer’s comment.

3) I strongly advise a thorough language revision for your text by an expert.

Response: We have invited Samantha C Karunarathna (samantha@mail.kib.ac.cn) to help us to polish it.

This manuscript is a resubmission of an earlier submission. The following is a list of the peer review reports and author responses from that submission.